# OpenReview forum: "In-Context Learning in Large Language Models: A Neuroscience-inspired Analysis of Representations"
_ICLR.cc/2024/Conference — Submitted to ICLR 2024_

### Official Review · Reviewer_8pv7 · 2023-10-27

**Soundness:** 2 fair
**Presentation:** 2 fair
**Contribution:** 1 poor
**Rating:** 3
**Confidence:** 4

**Summary:**

This paper purports to uncover how LLMs (i.e., their embedding and attention) change given in-context learning. Representational similarity analysis is applied, inspired by neuroscience. Experiments are run on Vicuna and Llama, including across a few investigations, including persona injection for jailbreaking.

**Strengths:**

- The combination of tasks is interesting, in this somewhat important (and hence somewhat crowded) area of research. It is likewise interesting to inject a persona _towards_ the latent representations, as jailbreaking and adversarial prompting is also a popular emerging area.
- The appendices are detailed and informative.

**Weaknesses:**

- Some rationale for using RSA was a bit shaky. In its introduction, RSA is touted as overcoming parametrized probing methods, but this is a bit of a ‘red herring’, as there are already (increasingly many) varieties of parameter-free probing methods (e.g., https://aclanthology.org/2020.acl-main.383.pdf). That’s not to say that the parameter-free nature of RSA isn’t a positive, but it doesn’t really highlight why we should use RSA (or any neuroscience-based approach) in particular.
- This paper does not seem to provide a novel contribution, as there is already plenty of work that describes ICL for LLMs. The only contribution seems to be the application of RSA however 1) the related work section reveals that RSA has already been variously applied elsewhere in NLP (so the magnitude of its contribution is quite diminished), and the method itself is not adequately described. Although the methodology of RSA can be inferred through Section 2, it should be defined quite explicitly, if this is the main contribution.
- Although the various errors in writing do not detract from comprehension, most of the time, it still casts doubt on the authors’ ability to attend to detail.
- The text in the figures are fairly unreadable, due to size, but perhaps this is unavoidable. Some of their important parameters are not discussed (e.g., Fig 3 is merely described as ‘behavioral error’ but the nature of the plots are not given)

**Questions:**

- Can you please run a spell-check over your paper (e.g., ‘anayzing’, 'appraoch’, etc) and also check the grammar (e.g., “RSA avoid this risk’, ‘the models behavior’, ‘to reduces the effect’) and LaTeX errors (like ?? instead of references)?
- Sec 2.4 — why was only the final layer of the LLM considered? How are the subsets a, b, and c determined? Why is ‘a’ not included in the computation of A(a,b,c)_x? Does this only apply to question-answering, as the second sentence of the second paragraph suggests?
- Sec 5 Why did you precede the instructions to be deceitful with instructions to be truthful? Would a fourth condition jump straight to instructions to be deceitful?
- Why did you give only incomplete references (i.e., without venue names) for at least 13 papers?

---

> ### Author Response · Authors · 2023-11-21
> **Author response to reviewer 8pv7 -- Part 1**
>
> **Benefits of RSA:** We agree that the parameter-free nature of RSA is not its only advantage, and that other parameter-free techniques exist. But RSA provides unique benefits beyond being parameter-free that make it well-suited as a tool for our goals in the paper. RSA allows direct comparison of the model's latent representational geometry to hypothesized relational structures specified a priori. This can be thought of as representational alignment between the representations and the ideal structure of the task. While probing can assess whether certain information is encoded in the representations, RSA provides a more holistic view of how well the entire relational structure of embeddings aligns with what an appropriate task representation should contain. In a nutshell, we think some of the key advantages of RSA are:
>
> - Allowing direct comparison of model representational geometry to hypothesized structures
> - Providing a holistic characterization of representational geometry
> - Avoiding complex modeling assumptions / risk of spurious correlations associated with parametric probing
>
> **Significance of our contribution:** Regarding the final point about RSA having been applied in various NLP contexts previously, we do acknowledge that it's not a wholly novel technique in NLP. But as we discuss in the related works section, this prior work has focused on small-scale models. To our knowledge, our work is the first to demonstrate the viability of leveraging RSA to analyze emergent behaviors in large state-of-the-art language models. Given the rapidly increasing scale and complexity of LLMs, developing interpretability tools that scale efficiently is an important open challenge. By scaling up RSA to analyze a 70B parameter model, we aim to demonstrate RSA's promise as a tool for characterizing complex behaviors in large modern LLMs. And of course, no one else has specifically investigated ICL with RSA.
>
> Please also refer to these new figures we created to visually describe our main contribution:
> - Overview of methods: https://figshare.com/s/1b8983f60c9f81075c32
> - Task and experiment design: https://figshare.com/s/e96a108f58e1ed381619
> -  Attention ratio analysis (ARA): https://figshare.com/s/9541214a2a8fbd37ba83
>
> **Definition of behavioral error:** We have included the details in figure captions. For example, in Fig 3, the caption describes the y axes as “absolute error for regression task” which is defined in equation (1). We have further clarified this in all figure captions, so it is easier to notice. We titled our y-axes as “behavioral error” so emphasize that in each task, we study behavioral changes that correlate with representation changes due to ICL.
>
> **The role of ‘a’ in $A(a,b,c)_x$:** The input parameter $a$ which represents a substring of $x$ is referenced by the index $i$ in $A_i,j$ and $A_i,k$. As described in Section 2.4, $i$ indicates token indexes of substring $a$ after tokenization of string $x$. This means that $a$ is used in summations for both the numerator and denominator when adding the attention values for all pairs of tokens between $a$ and $b$ as well as $a$ and $c$.
>
> **Determining the prompt substrings $a$, $b$, and $c$:** We designed the reading comprehension tasks so that these substrings are well defined. Each prompt has informative and distracting components that clearly define these substrings. The last paragraph of section 3 describes these components for the reading comprehension task.
>
> **Incomplete references and typos:** We apologize for our errors and thank the reviewer for bringing them to our attention. We have fixed all typos and Latex errors, and updated our references to include all information.
>
> **Layer choices:** We agree with the reviewer that investigating multiple layers and changes in representations across them is highly interesting. We experimented with the first, middle, and last layer before submission, and have now added more experiments to Appendix Sections C2 (Figures 17 and 18) and B2 (Figure 9) to show results for other layers of Llama-2 and Vicuna. Our results show that the middle layer and the last two layers demonstrate the same representational changes aligned with our hypothesis in the linear regression and persona injection tasks. The first layer of the network, however, does not show the same pattern. This is expected because the earlier layers in language models are known to reflect syntactical information as opposed to high-level task information. This observation applies to both models. We are happy to include more such experiments if the reviewer sees fit.
>
> **Continued in part 2.**

---

> ### Author Response · Authors · 2023-11-21
> **Author response to reviewer 8pv7 -- Part 2**
>
> **Persona injection prompt design:** We acknowledge the reviewer’s point that a prompt which directly instructs the model to lie is the most straightforward and would be the ideal experiment design. We initially conducted this experiment; however, Llama2 would output an end-of-sequence token immediately which prevented analysis of the model’s response with relation to the input prompt. We therefore attempted a succinct prompt injection which mirrors a simple adversarial attack. The prompt provided was the most succinct and simplistic adversarial prompt we found which changed the model’s response for both Vicuna 1.3 and Llama2.
>
> **Applications of Attention Ration Analysis:** The approach of analyzing attention ratios may have applicability beyond question and answer. It can be applied to identify potential hallucinations for any continuation assuming the model’s response could be decomposed into a claim and support structure. If a strong correlation exists, as the one found in our experiment, this threshold could serve as a measure to assist with prompt engineering. This would be done by finding phrasings of a question which maintained the threshold within a certain range based on observed values and their relationship to a desired property of the output.
>
> Thank you once again for your thoughtful review and detailed comments. We hope that our response addresses your concerns.

---

### Official Review · Reviewer_Sz99 · 2023-10-27

**Soundness:** 3 good
**Presentation:** 3 good
**Contribution:** 2 fair
**Rating:** 6
**Confidence:** 3

**Summary:**

The authors use several representation analysis methods (e.g., RSA, decoding, and attention metrics) to measure how LLM embeddings change based on examples provided in in-context learning (ICL).

One contribution, ancillary to the main analysis, is the "attention ratio" metric, which seeks to measure the weight of attention on relevant vs. irrelevant parts of an input.

The primary findings are based on experiments with the Llama2 and Vicuna models, where, for both models, the authors find that 1) adding ICL improves performance and 2) per several representation metrics, adding ICL seems to focus embeddings more on relevant parts of the prompt.

**Strengths:**

## Originality
This work is somewhat original in analyzing how embeddings change as ICL is used for models. The attention ratio metric is certainly new, and in general I like the idea of using these embedding measures to understand how a model react to different inputs beyond simple behavioral measures.

I note that the authors sometimes seem to include general behavioral changes due to ICL as one of their contributions (e.g., the first bullet point in the introduction section). There is already extensive work discussing the behavioral effects of ICL, some of which the authors cite, so I do not view this as one of the main contributions of this work

## Quality
The work itself seems largely well done, although aspect of the paper could be improved.

## Clarity
Overall, I found the paper quite clear overall, although there were some specific details or sections that were unclear (see weaknesses).

## Significance
I think if the authors further focused on representational, as opposed to behavioral measures, this work could be reasonably significant. As it stands, I think the work is somewhat torn between to thrusts, which weakens the overall message.

**Weaknesses:**

## Focus on behavioral vs. representational measures.
Perhaps my primary concern with this work is that it seems to focus quite a lot on presenting measures of behavioral changes due to ICL instead of representational changes. I believe the behavioral benefits of ICL have already been established (if not, please indicate what is new about these findings). Therefore, the exciting part of this work is the link between representational metrics and changes in inputs (and perhaps links between representational and behavioral changes). Many of the main figures, however, focus only on behavioral changes (Figure 3, Figure 4 a and b). Some figures are great (e.g., Figure 4 c), but I wish more were like that.

## Interpreting attention values
I am somewhat hesitant to use attention values in interpretability work (see "Attention is not Explanation" by Jain and Wallace). Can the authors justify using attention measures to understand how the model is behaving?

## Appendices:
The authors included many further results in appendices, which is overall a practice I encourage, but the appendices could benefit from refocusing and better presentation. Appendix B.1, for example, is very unclear to me.

## Minor:
1. Many of the violin plots used end up plotting uninterpretable extrapolations. For example, in Figure 3, by definition, no value can be less than 0, but the plots do go into negative values. The raw data are fine, but the plots seem confusing at first.

2. Figure 5 a-c would really benefit from some sorts of labels to say what the axes are. I believe I understand this from examples in the appendix (e.g., Figure 7), but it should be very clear in the main paper.

3. Figure 12 in the appendix seems wrong to me, especially because there are no labels on the axes. In particular, c) is supposed to represent the average of a and b, but a and b look the same.

**Questions:**

I only have a simple clarifying question: could the authors elaborate on how they "standardize the embeddings before computing pairwise cosine similarities" (section 2.1)?

---

> ### Author Response · Authors · 2023-11-21
> **Author response to reviewer Sz99**
>
> We thank the reviewer for allowing us to clarify our work. Our focus is indeed on a) **measuring changes in representation** following ICL (as well as changes in alignment of representations with what would be expected of the task structures), and b) **measuring and quantifying changes in attention** following ICL, and c) **quantifying** how these representation and attention changes correlate with behavioral changes. In other words, our goal is to ground behavioral changes in representational changes. The reviewer is correct that having behavioral tasks that show an effect with ICL is not a novel contribution in itself. However, our specific focus requires tasks that satisfy specific design principles: have components that enabled us to hypothesize about the representational similarity structures, display similar behavioral changes for both Llama 70B and vicuna13B. **Figures 1c, 1d, all of figures 2, 4, 5 and figures 6 c-h** focus on representational changes. For example, figures 4a and 4b show the accuracy of a probing classifier trained on the models’ representation to predict the slope of the line from the prompt embeddings but from your comment it sounds like you have interpreted that as a behavioral result. We hope this improves clarity and are happy to update the paper accordingly.
>
> Similarly to neuroscience, studying representations requires designing tasks with specific representational relations across different components. Thus, coming up with careful hypothesis-driven tasks was a crucial step: we had to carefully design tasks that have meaningful components, and demonstrate behavioral improvements due to ICL in these two specific models. With these tasks, we can study representational changes related to task components and how they correlate with behavioral changes.
>
> **Attention is not explanation:** We use attention in addition to model embeddings to measure representational changes, based on the hypothesis that token-level representational information may be lost in our choice of aggregation of token embeddings. Examining attention allows us to observe if there is a change in attention between specific parts of the prompt, and we do not mean to imply this is an explanation. For example, we would have reached similar conclusions even if the attention ratio defined in section 2 was always lower for truthful behavior than deceptive behavior. As stated in the paper in section 5, the attention values corresponding to different behaviors do exhibit a statistically significant difference in distributions.
>
> **Presentation issues:**
>
> - Violin plots: Thank you for pointing this out. All violin plots are fixed to not estimate negative values when representing non-zero distributions. We apologize if this caused any confusion.
>
> - Appendix Figure 12 and 5 a-c: We apologize for any confusion that the lack of labels may have caused. Including the prompts or even just the prompt IDs in the prompt similarity matrices would not yield a legible figure given the space constraints, but we have added clarification to the figure caption. To summarize, in this figure, each subfigure uses a distinct ordering of prompts aligned with the hypothesis they represent, so it is not visually perceptible that subfigure c is the average of the name and activity hypothesis.
>
> **Standardizing embeddings:** Similar to Timkey et. al (cited in section 2.1) we noticed that most prompt embeddings were dominated by one extremely large dimension pushing all similarities to high values. To mitigate the effect of such *rogue dimensions*, Timkey et al propose several methods including simply standardizing the embeddings before calculating cosine similarities.
>
> Thank you once again for your thoughtful review and detailed comments. We hope that our response addresses your concerns.

---

> > ### Comment · Reviewer_Sz99 · 2023-11-22
> >
> > Thank you for the response. Indeed, I had misread figures 4a and b, for example, as behavioral metrics rather than embedding classifier accuracy.
> >
> > I have increased my score to a 6.

---

### Official Review · Reviewer_DTAM · 2023-11-02

**Soundness:** 1 poor
**Presentation:** 2 fair
**Contribution:** 2 fair
**Rating:** 3
**Confidence:** 4

**Summary:**

This work aims to understand the effectiveness of in-context learning. Motivated by neuroscience research, this work proposed to study the latent representations of prompts and based on the similarity analysis to interpret how in-context examples impact model inference.

**Strengths:**

- The research problem is critical and challenging. Understanding in-context learning will not only help us understand the capacity of LLMs, but also help design better context or select better in-context examples for better using LLMs.
- The adopted approach, inspired by neuroscience, is interesting. Although, I do have some concerns about the actual implementation of the approach (explained in the following)
- The proposed work was supported by three very different language related tasks.

**Weaknesses:**

- The design of the method seems to be a little bit arbitrary. Not sure I understand the underly principle of design choices. For example
    - in section 2.1, the way of composing $n$ embeddings to one single embedding is to use max-pooling. Although I agree that max-pooling is one classical and popular method on representation composition, but I would like to know how this method was selected and to what extent this selection will impact the final conclusion
    - in section 2.1, it says “the most relevant subset of tokens representing important components of the prompt are considered”, by manually (I assume) selecting the “most relevant” subset, I am not sure whether the results can be used to represent LLM’s behavior
    - in section 2.2, it says “We designed a set of tasks with common components, such that representing those components is crucial in solving tasks”, if these components are crucial, it may be a good idea to talk a little bit more about these components, such as how these components are designed, and how do we know the combination of these components can be used to represent a way of solving tasks?
    - in section 2.4, why the attention of the final layer?
- Some (minor) writing issues, for example
    - A typo in the first sentence of section 2.3
    - The notions in section 2.4 are confusing, for example, both $a$ and $p$ represent prompts (?)
- Some issues about experiment design. For example,
    - Why choosing these three tasks, particularly the last time, rather than some tasks that in-context learning has been widely used?
    - What the role of RSA in section 3, which is not clear (it’s likely I may miss something important there)
    - About section 5, I am not sure whether the goal of this task is to study the robustness of the model or understand in-context learning is used for model inference. It’s an interesting task, even though I am a little skeptical about the application scenario. The most important question here is I am not sure how this experiment can help demonstrate the impact of in-context learning.

**Questions:**

Please refer to the previous section

---

> ### Author Response · Authors · 2023-11-21
> **Author response to reviewer DTAM -- Part 1**
>
> We thank the reviewer for their comments about our design decisions. We have added new experiments to the Appendix Sections C2 (Figures 17 and 18) and B2 (Figure 9) to address some of the concerns and we clarify others here.
>
> **Token choice for embeddings:** The tokens for embedding analyses were selected to represent the task portion of the prompt, excluding the ICL example tokens. In this way, the same tokens’ embeddings are considered from the prompt whether we are looking at a prompt with and or a prompt without ICL. It is not as manual a process since the question tokens are discoverable using the tokenizer and it is reasonable to assume that in any application scenario, the question/task tokens are distinguishable from the ICL example tokens. we have improved the clarity of our writing to more directly describe the token selection.
>
> **Task descriptions:** Each task is described in detail in its dedicated section (section 3, 4, and 5). For example, the reading comprehension task in section 3 is described to have informative and distracting components (sentences), and representing the informative sentence is crucial to solving the task because it contains the answer to the reading comprehension question.
>
> **Task design principles:** Similar to neuroscience, studying representations requires designing tasks with specific representational relations across different components. To identify such tasks, it is not sufficient to find tasks that show improvements with ICL, as what is needed are tasks with components that lead to representational similarities across different variations, about which we can make representational hypotheses. For instance, the reading comprehension task (Sections 3 and 5) allows us to vary *names* and *activities* separately and have hypotheses about the similarity of representations across multiple prompts with the same name, or the same activity. The same holds for multiple points that fall on the same line (Section 4), leading to the same slope. After successful ICL, one would expect improvements in *representational similarity or alignment with the hypothesis*.
>
> Thus, coming up with careful hypothesis-driven tasks was a crucial step: we had to carefully design tasks with meaningful components. This allowed us to have hypotheses about representations related to task components, continuous measures of improvement (e.g. similarity) after ICL, and how changes in representations and *changes in representational alignment with the hypothesis* correlate with behavioral changes following ICL.
>
> Another constraint was identifying tasks that demonstrate an ICL-induced behavioral change for both models. We also wanted a set of tasks which covered both numeric and verbal reasoning. Given the vast discrepancy in model capacity (13B vs 70B parameters) and the desire to reduce noise in our experiment, we chose three tasks which were self-contained, relatively succinct, and elicited change in behavior with ICL.
>
> We are open to the reviewer’s suggestion for running other/more prevalent ICL tasks that can satisfy the above design criteria if they could kindly share their thoughts with us.
>
> **Layer choice:**  In (https://aclanthology.org/P19-1356.pdf), Jawahar et al. used probing to demonstrate that earlier layers tended to capture syntactic information while later layers encode semantic information. Since we were interested in whether the model embeddings demonstrated a change in semantic encoding associated with the behavioral change, we focused on probing the last layer.
>
> We agree with the reviewer that investigating multiple layers and changes in representations across them is highly interesting. **We experimented with the first, middle, and last layer before submission, and have now added more experiments to Appendix Sections C2 (Figures 17 and 18) and B2 (Figure 9)** to show results for other layers of Llama-2 and Vicuna. Our results show that the middle layer and the last two layers demonstrate the same representational changes aligned with our hypothesis in the linear regression and persona injection tasks. The first layer of the network, however, does not show the same pattern. This is expected because the earlier layers in language models are known to reflect syntactical information as opposed to high-level task information. This observation applies to both models. We are happy to include more such experiments if the reviewer sees fit.
>
> **Continued in part 2.**

---

> ### Author Response · Authors · 2023-11-21
> **Author response to reviewer DTAM -- Part 2**
>
> **Embedding aggregation by max-pooling:** To address your question, we have included our analysis with mean pooling instead of max-pooling in the Appendix sections B2 (Figure 9), and C2 (Figures 17 and 18).
>
> In the regression task, mean pooling does not change the results for the middle and last layers, but the first layer’s representational correlation with our hypothesis seems to slightly increase with ICL when we aggregate the token embedding using mean-pooling instead of max-pooling. This observation applies to both models in both.
>
> In the persona injection task, mean-pooling, had a minimal impact on the correlation with the hypothesis matrices for all layers. All max pooled first layer token embeddings for 0-ICL produced the same encoding which led to a correlation of zero for all hypotheses. This is because a small subset of shared tokens, such as the trailing question mark and ‘what does’, collectively determined the elementwise max. We found that the middle layer and last layers had similar and high correlation for mean-pooled token embeddings of the prompt with both the activity and name hypothesis matrices. The correlation for both the middle and final layer increased when providing an ICL example for the name hypothesis for both models. Excluding the first layer, the inclusion of an ICL example improved correlation for a majority of model, layer, aggregation, and prompt type combinations.
>
> We are open to the reviewer’s suggestions in addition to this experiment.
>
> **The role of RSA is section 3:** In section 3 we use probing classifiers and attention ratio analysis. RSA is not used in section 3, but the task in section 5 is based on the task in section 3 and there we use RSA with name and activity hypothesis matrices.
>
> **ICL in section 5:** The intent of section 5 is to a. exhibit that providing a single ICL example improved both models’ abilities to respond truthfully (Figure 6 a-b) and b. to measure representational changes underlying the behavioral changes (Figure 6 c-h). We hope this clarifies how this section fits within the rest of the paper in that it does study the role of ICL in both behavioral and representational changes, and we are open to further suggestions by the reviewer.
>
> Thank you once again for your thoughtful review and detailed comments. We hope that our response addresses your concerns.

---

### Official Review · Reviewer_FbLG · 2023-11-03

**Soundness:** 3 good
**Presentation:** 2 fair
**Contribution:** 2 fair
**Rating:** 3
**Confidence:** 4

**Summary:**

This work aims to study the changes in a large language model that are brought about by in context learning (ICL). The authors study the changes in representation and attention in the last layer of 2 open source LLMs (Llama2 and Vicuna) in three toy tasks: answering a simple question in the presence of a distractor, linear regression, and susceptibility to adversarial attacks due to injecting of a deceitful persona. A variety of methods are used for this: probing classifiers, attention quantification, and representational similarity analysis (which is an approach borrowed from neuroscience). The results are that ICL helps the models improve their performance on the toy tasks, and that the changes in representation and attention in the last layer agree with those improvements.

**Strengths:**

- understanding ICL is a very interesting problem that is relevant to the ICLR community
- a variety of tasks and approaches are used for analyses
- two open source models are investigated

**Weaknesses:**

The research question is interesting and there is a nice start to investigate it here. However, currently the manuscript feels like it was put together hastily and like there was a breadth-first approach into several distinct directions and analysis techniques that can benefit from a more focused investigation.

My main concern is that it’s not clear what new insights are being offered here. The representations and attention are being studied at the last layer. We see that the behavior improves after ICL. The behavior is a direct function of the representations and attention at the last layer. What do we expect to have happened to those representations and attention if the behavior has improved? Is it possible for the representations and attention to not reflect that improvement? It would be more insightful if the authors investigate earlier layers and reveal what kind of effect ICL has on those representations.

Details of the models used are missing. Are they instruction-tuned? If so, on what datasets are they instruction-tuned?

Many typos, grammatical errors, and some citet/citep confusion.
Examples of errors: sec 2.3 “mebeddings”, page 4 “Section ?? for t-test results”, page 4 “has two distinguish”, page 4 “reading a book.”., page 6 “Hanna” without a last h, page 8 “to test this, designed”, page 8 “tested on used open source”
Examples of citet/citep confusion: 3rd paragraph on page 1, penultimate par on page 6, penultimate par on page 9

**Questions:**

One part of the paper which seems to have an unexpected result is in the last investigation of how ICL affects the representations after adversarial attacks related to injecting a deceitful persona. The unexpected result is that ICL makes the representations closer to the hypothesis model for names rather than actions, even though the answer should be an action. Can the authors offer their thoughts into why this is the case? Also, it would be helpful to clarify what is considered as “correct” response under these attacks: is it the response that aligns with the truth, or the response that aligns with the actual instructions that the model is supposed to follow?

More of a comment:
The inclusion of RSA is interesting. The field of interpretability already uses CKA a lot to examine representations and my first instinct when I read about RSA in the introduction was why RSA and not CKA. Later in the paper, I understood that what the authors really take away from RSA is the comparison of the representational space to a hypothesized model space. This should be made more obvious earlier on in the paper. I’m also not sold on the idea that you cannot get the same insights via probing (e.g. showing that decoding performance increases after ICL).

---

> ### Author Response · Authors · 2023-11-21
> **Author response to reviewer FbLG**
>
> We thank the reviewer for appreciating the premise of our work, that latent layers of LLMs should show changes with ICL that are reflected in behavior and allowing us to elaborate on our specific contributions. Our contributions are threefold:
> 1) We carefully design tasks that allow us to investigate what the structure of latent representations should be (hypothesis driven design). This allows us to ask questions like what the reviewer suggests “What do we expect to have happened to those representations and attention if the behavior has improved?”. As such, the question here is not whether it is “possible for the representations and attention to not reflect that improvement?” as the reviewer asks, but *how to measure* what these representational and attentional changes should look like, and what methods enable us to *measure* the changes.
>
> 2) We propose using a tried and tested method from neuroscience, *RSA* and correlation with a hypothesis matrix, to test the extent to which a representational structure (measured by RSA) becomes more similar to a hypothesized structure (correlation with a hypothesis matrix) following ICL. We can then measure the extent to which gradual changes in this correspondence lead to changes in behavioral error. As with neuroscience studies, it is not enough to show changes in representation but quantifiably *how ICL changes representation, and how these changes correspond with changes in behavior*. Another significance of this approach is that it allows us to decode answers to certain prompts directly from the representations, even if they were not reflected in *behavior*. This is an impactful finding, allowing future uses of open LLMs, in which some information can be directly captured from the layers.
>
> 3) We propose *ARA (Attention Ratio Analysis)*, a novel method to *measure how* attention changes following ICL. As above, we also investigate *how changes in attention following ICL correspond with changes in behavior*
>
> **Layer choice:** We agree with the reviewer that investigating multiple layers and changes in representations across them is highly interesting. We experimented with the first, middle, and last layer before submission, and have now added more experiments to Appendix Sections C2 (Figures 17 and 18) and B2 (Figure 9) to show results for other layers of Llama-2 and Vicuna. Our results show that the middle layer and the last two layers demonstrate the same representational changes aligned with our hypothesis in the linear regression and persona injection tasks. The first layer of the network, however, does not show the same pattern. This is expected because the earlier layers in language models are known to reflect syntactical information as opposed to high-level task information. This observation applies to both models. We are happy to include more such experiments if the reviewer sees fit.
>
> **Details of the models**: We chose two open-source decoder models from HuggingFace. Both models leveraged instruction fine tuning.
>
> 1. Vicuna v1.3 is fine-tuned from LLaMA with supervised instruction fine-tuning. The training data is around 125K conversations collected from ShareGPT.com.
>
> 2. Llama 2 was pretrained on 2 trillion tokens of data from publicly available sources. The fine-tuning data includes publicly available instruction datasets, as well as over one million new human-annotated examples.
>
> **Typos and Latex errors**: We apologize for our errors and thank the reviewer for bringing them to our attention. We have fixed all typos Latex errors.
>
> **Definition of “correct” answer in the persona injection task**: For evaluating the persona injection task, all results measure the model's ability to generate the truthful response. When asked to deceive the expectation is that the model should follow the earlier instructions to tell the truth and ignore instructions to the contrary.
>
> **The role of RSA:** RSA allows us to directly compare the representational geometry to a hypothesized model space. Specifically, RSA involves constructing representational similarity matrices that capture the pairwise similarities of instances in a representation, and then comparing these RSMs to an a priori similarity matrix that encodes some hypothesized relational structure. So, one of the notable benefits of RSA is to provide a simple and intuitive way to test correspondence to relational hypotheses. The reviewer's final point about probing sidesteps the various issues with parametric approaches mentioned in the paper. Although we use probing as well in the paper, we would like to emphasize RSA's ability to provide a holistic characterization of the representational geometry changes, rather than just improving decodability of one variable.
>
> Thank you once again for your thoughtful review and detailed comments. We hope that our response addresses your concerns.

---

> > ### Comment · Reviewer_FbLG · 2023-12-04
> >
> > Thanks for the response. The experiments with additional layers are a good start. Regarding the claim that "the earlier layers in language models are known to reflect syntactical information as opposed to high-level task information" -- there are different thoughts on this, with some research showing that syntactic info is encoded in middle layers rather than early layers (Jawahar et al. 2019 ACL).
> > I don't see a response to my question regarding offering insights into why, in the deceitful persona condition, the ICL makes the representations closer to the hypothesis model for names rather than actions. Including more discussion in an updated manuscript would be helpful.
> > My opinion of the paper in its current form still stands. If the central question of the work is studied in a bit more depth, then I can see this being an impactful contribution.

---

> ### Author Response · Authors · 2023-11-21
>
> Please also refer to these new figures we created to visually describe our main contribution:
>
> - Overview of methods: https://figshare.com/s/1b8983f60c9f81075c32
> - Task and experiment design: https://figshare.com/s/e96a108f58e1ed381619
> - Attention ratio analysis (ARA): https://figshare.com/s/9541214a2a8fbd37ba83

---

### Author Response · Authors · 2023-11-21
**Author response to all reviewers**

We would like to thank all the reviewers for their insightful comments and suggestions. We appreciate that the reviewers recognize our paper as addressing a ‘critical and challenging’ (DTAM) problem that is ‘relevant to the ICLR community’(FbLG) through ‘a variety of tasks and approaches’ (FbLG). We hope that our responses and paper updates alleviate the concerns raised particularly about the clarity and quality of the presentation of the paper which was a common theme among reviewers. Based on your collective feedback, we have updated the draft in the following ways:

- We added experiments to the appendix sections B and C to show results for various layers of the models in addition to the last layer.
- We added experiments to the appendix sections B and C to compare max-pooling and mean-pooling token embedding aggregation.
- We have addressed all issues regarding the presentation of the paper, including fixing typos and Latex errors, improving our Figures, and the general clarity of the writing.
- We have created additional Figures to further clarify our proposed method:
  - Overview of methods: https://figshare.com/s/1b8983f60c9f81075c32
  - Task and experiment design: https://figshare.com/s/e96a108f58e1ed381619
  - Attention ratio analysis (ARA): https://figshare.com/s/9541214a2a8fbd37ba83

Please find our response to each individual reviewer below the reviewer comment.

---

### Meta-Review · Area_Chair_wNrg · 2023-12-06

**Metareview:**

This work aims to study the changes in a large language model that are brought about by in context learning (ICL). The authors study the changes in representation and attention in the last layer of 2 open source LLMs (Llama2 and Vicuna) in three toy tasks: answering a simple question in the presence of a distractor, linear regression, and susceptibility to adversarial attacks due to injecting of a deceitful persona. A variety of methods are used for this: probing classifiers, attention quantification, and representational similarity analysis (which is an approach borrowed from neuroscience). The results are that ICL helps the models improve their performance on the toy tasks, and that the changes in representation and attention in the last layer agree with those improvements.

All reviewers agreed that this was an important problem that needed more studying, but questioned the methodology and whether the results truly informed the problem being addressed. All reviewers also agreed the paper could be clearer and more detailed, thereby benefitting from another round of review.

**Justification For Why Not Higher Score:**

See meta-review.

**Justification For Why Not Lower Score:**

N/A

---

### Decision · Program_Chairs · 2024-01-16

Reject